# Clonality and Persistence of Multiresistant Methicillin-Resistant Coagulase-Negative Staphylococci Isolated from the Staff of a University Veterinary Hospital

**DOI:** 10.3390/antibiotics11060811

**Published:** 2022-06-16

**Authors:** Joaquín Rey, María Gil, Javier Hermoso de Mendoza, Alfredo García, Gemma Gaitskell-Phillips, Carlos Bastidas-Caldes, Laura Zalama

**Affiliations:** 1Unidad de Patología Infecciosa y Epidemiología, Facultad de Veterinaria, Universidad de Extremadura, 10003 Cáceres, Spain; mariamilgm@unex.es (M.G.); jhermoso@unex.es (J.H.d.M.); laura.zalama94@gmail.com (L.Z.); 2Área de Producción Animal, CICYTEX-La Orden, 06187 Badajoz, Spain; alfredo.garcia@juntaex.es; 3Unidad de Reproducción y Obstetricia, Facultad de Veterinaria, Universidad de Extremadura, 10003 Cáceres, Spain; ggaitskell@unex.es; 4One Health Group, Facultad de Ingeniería y Ciencias Aplicadas (FICA), Biotecnología, Universidad de las Américas (UDLA), Quito 170125, Ecuador; cabastidasc@gmail.com

**Keywords:** methicillin-resistant, coagulase-negative staphylococci, antimicrobial resistance, multiresistant, clonality, persistence, SCC*mec*, PFGE

## Abstract

The aim of this study was to characterize methicillin-resistant coagulase-negative staphylococci (MRCoNS) isolates from the healthy staff of a university veterinary hospital in order to assess their importance as a reservoir of antimicrobial resistance and to determine their population structure and evolution. The study duration was over two years (2020–2021), 94 individuals were analyzed in duplicate, and 78 strains were obtained. The overall prevalence of methicillin-resistant strains detected throughout the study was 61.7%, with point prevalence values of 53.2% in 2020 and 31.5% in 2021. A total of 19.1% of the individuals analyzed were carriers throughout the study. The most frequently identified MRCoNs were *Staphylococcus epidermidis* (92.3%) and *S. warneri* (3.8%). A total of 75.6% of the isolates obtained showed the development of multi-resistance, preferentially against erythromycin, gentamicin, and tetracycline, and to a lesser extent against fusidic acid, norfloxacin, and clindamycin; these antimicrobials are frequently used in the veterinary field. Although most of the *S. epidermidis* isolates obtained showed wide genetic variability and low dispersion, which are characteristic of community-associated isolates, a small number of strains spread between individuals in close physical proximity and were maintained over time, forming stable clones. These clones generally maintained the same type of staphylococcal cassette chromosome *mec* (SCC*mec*) and had a similar antimicrobial resistance pattern.

## 1. Introduction

The genus *Staphylococcus* has classically been divided into two groups based on diagnostic criteria: coagulase-positive staphylococci (CoPS), which possess significant pathogenic potential and are responsible for serious clinical processes, including *Staphylococcus aureus* and *S. pseudintermedius* as the most representative species in humans and animals, respectively; and coagulase-negative staphylococci (CoNS), with a lower pathogenic potential and formed of a larger number of species [1]. CoNS represent a wide and heterogeneous group, consisting of 47 species and 23 subspecies that form part of the skin microbiota and mucous membranes in humans and animals [2]. Among the most common species in humans are *S. epidermidis*, *S. haemolyticus*, and *S. saprophyticus*. Although they have limited pathogenic potential, they can cause nosocomial infections, mainly in immunocompromised patients, normally related to implant devices such as catheters or prostheses, as well as various chronic infections. This characteristic is favored by their ability to form biofilms and interfere with the immune response [3]. However, the real importance of this group lies in their ability to accumulate antimicrobial resistance genes over time. These genes are located in plasmids, insertion sequences, and transposons, which are mobile structures easily transferable between bacteria, regardless of the pathogenic or non-pathogenic nature of their recipient [4]. The accumulation of resistance genes is particularly common in bacterial populations subjected to high selective pressure as a result of the frequent or continuous use of antimicrobials, as occurs in human and veterinary clinical environments. In these circumstances, CoNS accumulate antimicrobial resistance genes that can easily be transferred to other pathogenic bacteria, such as *S. aureus*, which allows integration of both their own virulence potential and the transferred resistance potential. A paradigmatic case of this phenomenon is that of methicillin resistance (MR) in *Staphylococcus* (MRS). MRS is related to the expression of a modified penicillin-binding protein PBP2a encoded by the *mecA* gene and located in a complex mobile structure known as staphylococcal cassette chromosome *mec* (SCCmec) present in the bacterial chromosome, which, in addition to conferring resistance to all β-lactams, often harbors other types of resistance. Although several studies have shown the high prevalence of MR in CoNS (MRCoNS) clinical isolates, with values frequently higher than those detected in *S. aureus* (MRSA), this resistance is not so frequent in community-associated isolates [2]. 

A notable and little recognized example of this is represented by staff in veterinary hospitals, who, due to their close contact with clinical cases in animals, may harbor asymptomatically transferred MRCoNS strains, frequently associated with other types of resistance, thus becoming an asymptomatic reservoir of these strains. This exchange of resistance between species is favored by the similarity of the mobile genetic elements coding for them [5].

Therefore, this study aimed to characterize the MRCoNS present in the staff of a university veterinary hospital, determine their prevalence, establish the associated antimicrobial resistance patterns, and determine the structure and persistence of the bacterial population involved, with the ultimate aim of determining the importance of these bacteria and the individuals that carry them as reservoirs of resistance over time in a highly selective hospital environment.

## 2. Results

### 2.1. Prevalence and Individual Persistence of mecA Gene

The *mecA* gene was detected in 50 individuals in the first sampling (53.2%) and in 28 in the second (31.5%), while 58 individuals were positive at some point during the study (61.7%). In 18 individuals there was a continuous carrier status (19.1%) (18 of 94). The total number of *mecA*-positive isolates obtained was 78 (Table 1). 

### 2.2. Species of MRS

Five different MRS species were isolated among the 78 *mecA*-positive isolates obtained: *S. epidermidis* (*n* = 72; 92.3%), *S. warneri* (*n* = 3; 3.8%), *S. haemolyticus* (*n* = 1; 1.3%), *S. pseudintermedius* (*n* = 1; 1.3%), and *S. sciuri* (*n* = 1; 1.3%). All isolates obtained were CoNS, with the exception of *S. pseudintermedius*, which was CoPS (Table 2). 

### 2.3. Antimicrobial Resistance

A total of 75.6% of the isolates obtained (59/78) showed multi-resistance: 67.9% (53/78) to 3–6 antimicrobials, and 7.7% (6/78) to more than 6 antimicrobials.

All isolates obtained showed resistance to penicillin (P), cefoxitin (FOX), and oxacillin (OX) as a general characteristic of MR, except the only *S. sciuri* isolate that was obtained, which was sensitive to cefoxitin and oxacillin. Resistance to β-lactams was concurrent with other resistances, with a frequent (>40% of isolates) association with erythromycin (E) (66.7%), gentamicin (CN) (57.7%), and tetracycline (T) (46.1%); medium (40–20% of isolates) with fusidic acid (FD) (29.5%), norfloxacin (NOR) (28.2%), clindamycin (DA) (25.6%), and trimethoprim/sulphamethoxazole (SXT) (21.8%); and low or zero (<20% of isolates) with rifampicin (RD) (3.9%), chloramphenicol (C) (2.6%), quinuspristin/dalfopristin (QD) (0%), linezolid (LNZ) (0%), and vancomycin (VA) (0%). Joint presentations of different phenotypic resistances (P-FOX-OX/CN/E: 33.3% and P-FOX-OX/CN/E/T: 23%) were frequent (Figure 1). 

### 2.4. Clonality and Individual Persistence of S. epidermidis

Macrorestriction profiles were obtained in 43 of the 48 isolates of *S. epidermidis* isolated in the first sampling, with 32 different restriction patterns or pulsotypes (PTs) that grouped strains considered as “closely related” (similarity ≥ 90%). Of these, 25 PTs (58.1%) include a single strain each (singletons), while the remainder (7 PTs) include between 2 and 4 strains: PTs 8, 10, 12, and 15 (2 strains each), PTs 11 and 25 (3 strains each), and PT 22 (4 strains). Five strains could not be typed (Figure 2, Table 3).

Of the 18 individuals considered as persistent carriers, 8 individuals (2, 3, 7, 8, 10, 11, 12, and 16) had “indistinguishable” isolates throughout the study (similarity = 100%) and 3 individuals (4, 13, and 14) harbored strains considered as “closely related”, while the remaining individuals (1, 5, 6, 9, 15, 17, and 18) had “unrelated” strains (Figure 3).

### 2.5. SCCmec Typing

Of the 78 strains investigated, 9 (11.5%), 20 (25.6%), 7 (9%), and 34 (43.6%) exhibited *mec* complex classes A, B, C1, and C2, respectively, while a known *mec* complex was not identified in 8 strains (10.3%). Concerning the *ccr* complex, 69 (88.5%) exhibited AB2 (65 independently and 4 associated with C1) and 5 (6.4%) exhibited C1 (1 independently and 4 associated with *ccr*AB2), while 8 (10,3%) isolates were non-typeable (NT). 

Only 30 strains (38.4%) could be assigned to a particular SCC*mec* type, including 20 (25.6%) to type IV, 9 (11.5%) to type II, and 1 to type V (1.3%). Of the remaining 48 strains, 40 (51.3%) had unusual *mec-ccr* combinations (UC) that could not be assigned by the classical SCC*mec* scheme, and 8 (10.3%) were non-typable (NT) as no known *mec* or *ccr* complex could be identified. Atypical *mec-ccr* complex combinations included 29 (37.2%) C2-AB2, 7 (9%) C1-AB2, and 4 (5.1%) C2-AB2/C1 (Table 4).

## 3. Discussion

The study of methicillin resistance has traditionally been focused on *S. aureus* (MRSA) as the responsible agent for severe clinical processes and due to the therapeutic difficulties it poses, with fewer studies related to CoNS, despite its growing involvement in nosocomial infections and its increasing importance as a reservoir of antimicrobial resistance (AR). In relation to the resistances present, CoNS can be divided into clinical isolates, those which have had previous contact with antibiotics, and community isolates, in which this contact has been scarce or non-existent [6]. Contact can result in the development of AR as an adaptive response to the presence of these antimicrobials. Consequently, CoNS clinical strains have experienced a marked increase in AR in recent years [7,8]. When focusing exclusively on MR, these percentages have surpassed those detected in MRSA, rising in recent decades from 11% to 55% [9], reaching 80–90% in some countries [10]. CoNS community strains, however, exhibit lower levels of resistance because they have had less contact with antimicrobials [11]. 

In this study, MR strains with a prevalence more typical of clinical isolates than of community isolates were obtained, although they were acquired from apparently healthy individuals with no previous antimicrobial treatment. Nevertheless, all people sampled had a direct link to clinical cases in the veterinary hospital setting. Thus, an overall prevalence of MR isolates of 61.7%, with a point prevalence of 53.2%, were obtained in the first sampling and 31.5% in the second, while the percentage of persistent carriers was 19.1%, values that were much higher than those detected in other professional groups [12]. The differences in prevalence observed throughout the study and the intermittency of the carrier character are open to different interpretations. Factors related to the individual themself, fundamentally of an immune nature [13]; factors intrinsic to the bacterium itself, related to its capacity for adherence and colonization [14]; and environmental factors, mainly related to humidity and temperature [15], may condition the survival of these bacteria over time. In relation to this last point, we would like to draw attention to the fact that the second round of sampling in our study, with a lower prevalence than the first, was carried out during one of the hottest months of the year, with the lowest humidity, factors that may have somehow conditioned the viability and persistence of the strains investigated. Other types of individual differences related to the type of animal or the degree of exposure were not considered relevant since they are mostly small animals handled in a comparable way in the different hospital services, so the degree of exposure is similar.

*Staphylococcus epidermidis* was the most frequently isolated MRCoNS in this study (72 isolates, 92.3%). This species, which is the most frequent and ubiquitous of all CoNS, is of great epidemiological and clinical importance as it is the main reservoir of AR for other staphylococci [16], and is responsible for the majority of nosocomial outbreaks [17]. Other MRCoNS have been isolated in smaller proportions and are of less epidemiological significance: *S. warneri* (3 isolates, 3.8%), *S. pseudintermedius* (1 isolate, 1.3%), *S. haemolyticus* (1 isolate, 1.3%), and *S. sciuri* (1 isolate, 1.3%). Two of the species mentioned deserve particular consideration: the human detection of *S. pseudintermedius* could indicate zoonotic dissemination as it is predominantly an animal species, and it was also the only coagulase-positive species detected; and the identification of *S. sciuri* as an MR may be erroneous as this species carries an *mecA* gene homologue, not encoded in the SCC*mec* but in the chromosomal DNA, which does not confer resistance to β-lactams [18]. In fact, this was the only isolate that in the subsequent resistance study was sensitive to this group of antibiotics.

In this study, MR was frequently associated with other types of resistance: 75.6% of isolates were multidrug-resistant, of which 33.3% and 23%, respectively, had the resistance associations P/FOX/OX, E, CN, and P/FOX/OX, E, CN, TE. The presence of multi-resistance is more common among clinical isolates than in community isolates [19,20], so the high levels detected in our study may indicate some kind of horizontal transfer from clinical animal strains. The type of resistance developed is conditioned by the frequency of use of each specific antibiotic. Thus, the most frequently detected resistances (P/FOX/OX, E, CN, and TE) in this study correspond to antimicrobials commonly used in veterinary clinics for diarrheal, pneumonic, or obstetric conditions [21]. The subsequent spread of these resistances from animal to human strains would be favored by the location of their coding genes in mobile elements of different natures (plasmids, transposons, SCC) [4], which are easily transferable structures between bacterial populations. 

Classically, determination of the clonal relatedness of *S. epidermidis* has been performed by the simultaneous use of two techniques: SCC*mec* typing and macro-restriction profiling by *SmaI* enzymatic digestion followed by pulsed-field gel electrophoresis (PFGE). Although neither technique is perfect, both techniques, used together, complement each other and provide optimal results [22]. 

Methicillin resistance in *Staphylococcus* is related to the acquisition of the *mecA* gene that encodes for an additional penicillin-binding protein, known as PBP2a, which has a low affinity for all b-lactams [23]. This gene is part of a mobile genetic element, known as the staphylococcal cassette chromosomal (SCC*mec*), which is easily transferable between staphylococci. It is composed of three basic structures: *mec* gene complex, which contains the *mec* gene itself (A, B, C, D, and/or E), its regulatory elements (*mec*CR1 and *mec*CI) and the associated insertion sequences (ISs); *ccr* gene complex, which encodes a site-specific recombinase (*ccr*AB and *ccr*C) that allows integration/excision of the SCC*mec* into the *Staphylococcus* chromosome; and the joining region J (J1, J2, and J3), which may contain additional virulence determinants/antimicrobial resistances [24]. The arrangement and genetic composition of these structures determine the different types of SCC*mec*. So far, 14 different types and subtypes have been described, each of them more frequently found in specific population groups and habitats [25].

While this classification is well suited to MRSA, it is more complicated to apply to MRCoNS, as structural components of their SCC*mec* may differ, being atypically rearranged, duplicated, or absent, in comparison to those present in MRSA [2], so that a large proportion of them cannot be typed. In fact, in this study only 30 strains (38.5%) of the 78 investigated could be assigned a specific SCC*mec* type, while the remaining strains (*n* = 48; 61.5%) either had unusual combinations in their *mec-ccr* complexes (*n* = 40, 51.3%), or were not typeable as no known *mec-ccr* complexes were detected (*n* = 8, 10.3%). Among the SCC*mec* types detected, type IV (*n* = 20; 25.6%) and type II (*n* = 9; 11.5%) stood out due to their frequency, while type V was only detected in one isolate (1.3%). 

SCC*mec* type IV is the most prevalent among the MR *S. epidermidis* [26], also being present in community-acquired MRSA (CA-MRSA) strains with which it has a high similarity (98–99%) [6,27]. This circumstance favors the exchange of this structure between both species, thus making *S. epidermidis*, due to its ubiquity and abundance, the main disseminating reservoir for *S. aureus* [28]. SCC*mec* type IV is structurally the smallest, and does not harbor any resistance genes other than *mec* [6]. Chronologically it was the first to appear [29] so it may have been the origin of the SCC complex for the rest of *Staphylococcus*. Regarding SCC*mec* type II, it is not as frequent among MRCoNS [6]. It is usually associated with clinical strains of hospital-associated MRSA (HA-MRSA) and also carries additional resistance genes [28], so its detection in this study seems exceptional as it has been detected in *S. epidermidis* (8 isolates) and *S. warneri* (1 isolate). Finally, type V was only detected in one strain of *S. epidermidis*. 

With regard to atypical associations, the most frequent finding was the *mec*C2-*ccr*AB2 combination (*n* = 29, 37.1%), and to a lesser extent *mec*C1-*ccr*AB2 (*n* = 7, 9%). Complex associations of a single *mec* complex (*mec*C2) with several *ccr* complexes (*ccr*AB2/C1) (*n* = 4; 5.1%) were also detected, associations previously described in *S. aureus*, *S. epidermidis*, and other CoNS [29]. The *ccr*AB2 complex was the most frequently detected (*n* = 69, 88.5%), both present individually or associated with *ccr*C. The abundance and variability of the recombinases detected (*ccr* complex) favors the exchange of resistance genes associated with them, both in relation to methicillin and other types of resistance [29]. 

In this study, broad genetic variability was found among the *S. epidermidis* isolates obtained. Thus, 43 isolates had 32 pulsotypes (PTs), and 25 isolates (58.1%) of the PTs obtained consisted of only 1 isolate (singleton). In this regard, it should be noted that *S. epidermidis* has high genetic diversity, much higher than that observed in other coagulase-negative staphylococci [22]. This fact is related to the rapid evolution of its chromosome due to the frequent exchange of mobile elements and to its high recombination rates [30]. This is evidenced by the presence in the species of homologous regions in different types of SCC*mec* and the high rate of insertion/excision of the IS56 region in *S. epidermidis* [31]. This plasticity gives the species a high capacity for variability and adaptation. Thus, the different clones of the species diverge over time by adaptation to the selective pressure exerted by various environmental factors, mainly that represented by the presence of antimicrobials. 

Despite this great genetic variability, in this study certain *S. epidermidis* clones were found to expand in individuals of certain groups while retaining the same SCC*mec* type and their resistance characteristics. This was observed with PTs 22 (4 isolates from group 5), 11 (3 isolates from group 7), and 25 (3 isolates from group 8), which include isolates that are genetically close to each other (similarity > 90%), with the same SCC*mec* type and similar phenotypic patterns of resistance. These clones not only became dominant but some of them also persisted for at least 6 months. This is the case for isolate numbers 138, 141, and 172 (group 5); 134 and 148 (group 7); and 165 (group 8). Therefore, all these isolates have a common origin, have been transferred horizontally between close individuals, and have maintained a certain genetic stability over time.

Regarding the evolution of the *S. epidermidis* strains in the 18 individuals considered as persistent carriers, the 8 individuals harboring strains considered as indistinguishable (100% similarity) maintained the same SCC*mec* type and an identical phenotypic resistance pattern with slight differences throughout the study. In the same way, the 3 individuals with closely related strains (similarity ≥ 90%) maintained the same resistance pattern, although two of them incorporated a complementary *ccr* complex (C1), while the remaining 7, with strains considered as unrelated (similarity < 90%), presented a greater diversity of SCC*mec* types and resistance patterns. 

Therefore, two different types of *S. epidermidis* population can be considered in our study: one, with a majority, which is heterogeneous, represented by strains that have little or no dispersal, evolve rapidly, and do not persist over time, which are features of community associated isolates [2,32]; and another, present only in some individuals, characterized by dissemination among individuals working in close proximity, that retain some genetic stability and persist over time (at least up to 6 months) adopting a clonal structure, which are typical features of clinical associated isolates [33]. The former, due to their abundance, variability, and adaptability, could be responsible for the transfer of the resistance observed between the animal and human spheres, with the repercussions that this behavior occurs in public health settings; and the latter, more stable and adapted, being responsible for the clinical processes produced.

## 4. Materials and Methods

### 4.1. Study Design and Population

Our work was an exploratory and cross-sectional study carried out among the teaching, care, and research staff of the clinical hospital of the Faculty of Veterinary Medicine in University of Extremadura, Cáceres (Spain). The study was carried out during the months of December 2020 and June 2021, in a location with climatic characteristics typical of a Mediterranean climate with Atlantic influences: temperate and humid in winter and hot and dry in summer. The individuals analyzed belonged to 8 different units or departments with direct contact with animals, mainly small animals suffering from medical, surgical, or obstetric conditions of varying etiologies. All the personnel were between 25 and 55 years old; both groups had a similar male to female ratio, did not report suffering from pathological processes at the time of sampling, and had not been treated with antimicrobials in the six months prior to sampling.

### 4.2. Sample Collection

Nasal swabs were obtained from a total of 94 individuals in two serial samplings 6 months apart (December 2020 and June 2021), obtaining a total of 183 swabs. Five individuals could not be sampled the second time as they were unavailable at the time of sampling. The swabs obtained were kept in Stuart-Amies preservation medium (DeltaLab, Madrid, Spain) and refrigerated (5 °C) until further processing, which was never more than 24 h after obtaining the sample.

### 4.3. Bacterial Culture Detection of the mecA Gene and Conservation

The swabs obtained were pre-enriched in brain heart infusion (BHI) (Oxoid, Madrid, Spain) supplemented with ClNa (6.5%) to enhance the selective growth of *Staphylococcus* [34] and incubated at 37 °C/24 h. Aliquots of 100 mL were streaked onto Columbia blood agar (Oxoid, Madrid, Spain) supplemented with cefoxitine (4 mg/L) and incubated under aerobic conditions at 37 °C/24 h. The presence of the *mecA* gene was investigated using PCR, first from the confluent of each plate, and subsequently, if positive, from the re-isolation of individual colonies. This procedure was repeated until a positive result was obtained in at least one colony.

DNA extraction was carried out by suspending each colony in 250 mL of sterile distilled water and heating the suspension at 95 °C/10 min. After centrifugation at 10,000 rpm for 5 min, the supernatant was used as a template. PCR amplification of the *mec*A gene has been described previously [35]. Briefly, it was carried out for a final volume of 25 mL containing 12.5 mL of 2X FastGene^®^ Optima HotStar (Nippon Genetics, Düren, Germany), primers mA1 (5′-TGCTATCCACCCTCAAACAGG-3′) and mA2 (AACGTTGTAACCACCCCAAGA) at a final concentration of 0.5 M of each primer, 8 mL of water, and 2 mL of template. PCR conditions were as follows: denaturation (95 °C/3 min), 30 cycles of denaturation (94 °C/30 s), annealing (57 °C/30 s), extension (72 °C/3 min), with a final elongation at 72 °C for 10 min. Amplification produced a band with a molecular size of 286 bp. Individual *mecA*-positive colonies were stored at −70 °C in cryopreservation spheres pending further analysis.

### 4.4. Species Identification

Matrix-assisted laser desorption/ionization time-of-flight mass spectrometry (MALDI-TOFF MS) [36] was used to identify all methicillin-resistant isolates obtained, whose Gram staining and biochemical characteristics (oxidase and catalase) were compatible with *Staphylococcus*, considering Brukers’ cut-off value for reliability (LogScore > 1.70).

### 4.5. Antimicrobial Resistance

A panel of 15 antimicrobials representing different classes was selected, all of them with a background history of antimicrobial resistance in the *Staphylococcus* genus. Antimicrobial susceptibility testing on 14 antimicrobials was performed using the disc diffusion method, while for vancomycin, the minimum inhibitory concentration (MIC) was used (Etest, Biomerieux, Spain). Disk diffusion methodology, MIC determination, and breakpoint tables were based on the recommendations set out by EUCAST (V. 9.0 and 10.0) [37]. The following discs (Oxoid, Madrid, Spain) were used: penicillin (P) (1 unit); cefoxitin (FOX) (30 mg); oxacillin (OX) (1 mg); tetracycline (TE) (30 mg); chloramphenicol (C) (30 mg); gentamicin (CN) (10 mg); erythromycin (E) (15 mg); clindamycin (DA) (2 mg); quinupristin/dalfopristin (QD) (15 mg); linezolid (LZD) (10 mg); rifampicin (RD) (5 mg); norfloxacin (NOR) (10 mg); trimethoprim/sulphamethoxazole (SXT) (1.25/23.75 mg); and fusidic acid (FD) (10 mg). *Staphylococcus aureus* ATCC 29213 and NCTC 12493 were used as control strains. An isolate was considered multi-drug resistant when it showed simultaneous resistance to three or more groups of antimicrobials [38].

### 4.6. Staphylococcal Cassette Chromosome mec (SCCmec) Typing

The protocol used was based on the methodology described by Yamaguchi (2020) [39] that allows differentiation of 11 different types of SCC*mec* (I-XI), and subtypes present in types II and IV. Its development is based on the use of 4 multiplex PCRs to identify each of the members of the SCC*mec*.: PCR-1 (*mec*A and *ccr* genes complex types); PCR-2 (*mec* gene complex classes A, B and C2); PCR-3 (ORFs in J1 region of type IV SCC*mec*); and PCR-4 (ORFs in J1 region of type II SCC*mec*). It also includes a uniplex for identifying other target *mec*C (Class C1). Strains NCTC10442 (type I), N315 (type II), 85/2082 (type III), JCSC4744 (type IVa), JCSC2172 (type IVb), JCSC4788 (type IVc), JCSC4469 (type IVd), WIS (type V), HDE288 (type VI), JCSC6082 (type VII), C10682 (type VIII), JCSC6943 (type IX), JCSC6945 (type X), and LGA251 (type XI) were used as control strains.

### 4.7. Phylogenetic Analysis Using Pulsed-Field Gel Electrophoresis (PFGE)

Firstly, we compared all *S. epidermidis* isolated from the first sampling (48 isolates), and secondly *S. epidermidis* obtained in both samplings from persistent carriers (36 isolates from 18 individuals), so effectively two studies were performed, at one point in time and one over a period of time. Identification of the genetic relationship between the isolates obtained was performed by macrorestriction with *SmaI* enzyme followed by pulsed-field gel electrophoresis (PFGE) (Chef-Mapper, Biorad^®^, Hercules, CA, USA) following the protocol described by PulseNet (CDC) [40]. The patterns generated were analyzed using InfoQuest^®^ software in conjunction with visual inspection using previously defined criteria [41] in which band patterns with 3 differences or fewer between strains are defined as “closely related” (similarity coefficient ≥90% using cluster analysis), and “indistinguishable” when there were no differences in the band patterns (similarity = 100%). A dendrogram was derived from the unweighted pair group method using the arithmetic average (UPGMA) and based on the Dice coefficient at a band optimization of 0.8% and 1.3% band position tolerance [11]. PFGE types were arbitrarily assigned as No. 01, 02,…, 32.

## 5. Conclusions

The majority of MRCoNS isolated from staff at the university veterinary hospital are *S. epidermidis* harboring, together with methicillin resistance and multiple resistances to antimicrobials frequently used in veterinary medicine. These resistances can be easily transferred between bacteria and species due to the presence of mobile genetic elements, such as SCC*mec* type IV, similar to that in *S. aureus*, SCC*mec* type II, and the abundance of specific recombinases (A2, B2, and C1).

Although most *S. epidermidis* strains show a broad genetic diversity, a small proportion of these strains expand and are maintained over time between individuals in the same environment, developing stable clones. Genetically, the evolution of these clones is minimal, generally retaining the same SCC*mec* and maintaining similar phenotypic resistance patterns. Therefore, from a structural point of view, two population types can be considered: a majority, composed of strains with low diffusion, high diversity, and low permanence, which are characteristics of community strains; and a minority population, composed of clones that are disseminated and persist for long periods of time, with characteristics that correspond to clinical strains. The possible zoonotic character of these strains and its genotypic profile of resistance and virulence should be confirmed in future studies. 

## Figures and Tables

**Figure 1 antibiotics-11-00811-f001:**
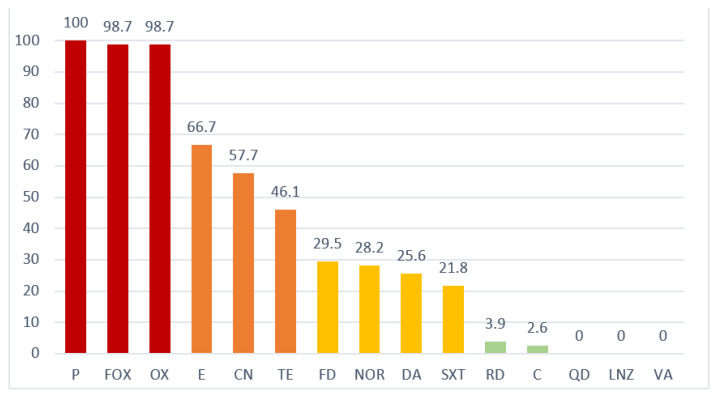
Percentage of Staphylococci resistant to different antimicrobials isolated from the staff of a university veterinary hospital between 2020 and 2021. P, penicillin; FOX, cefoxitin; OX, oxacillin; E, erythromycin; CN, gentamicin; TE, tetracycline; FD, fusidic acid; NOR, norfloxacin; DA, clindamycin; SXT, trimethoprim/sulphamethoxazole; RD, rifampicin; C, chloramphenicol; QD, quinuspristin/dalfopristin; LNZ, linezolid; VA, vancomycin.

**Figure 2 antibiotics-11-00811-f002:**
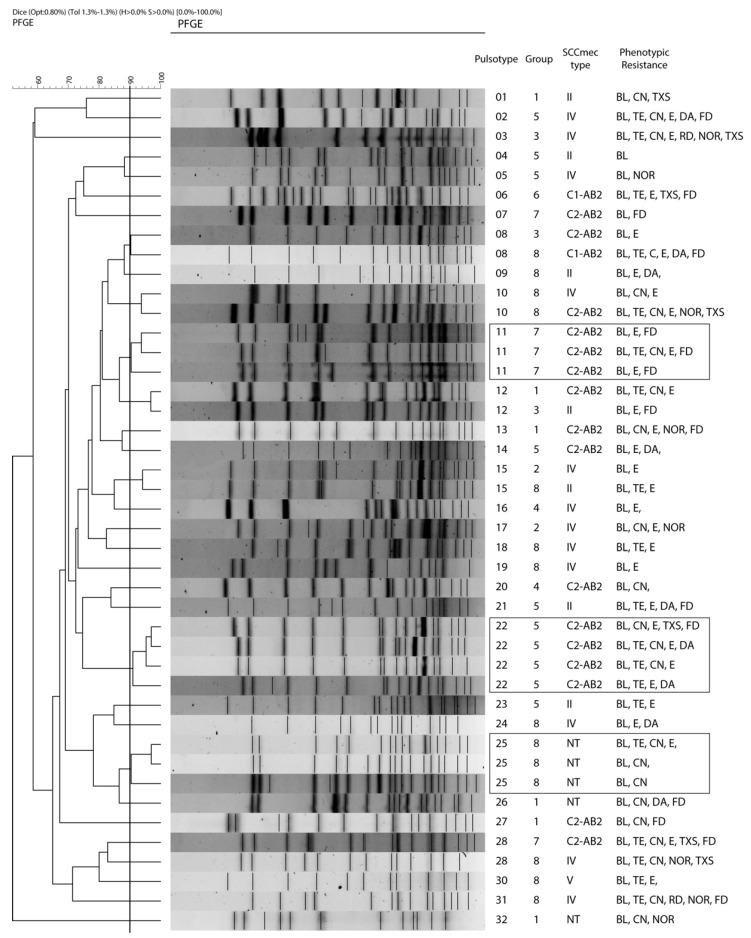
Dendrogram based on PFGE macrorestriction pattern of MR *S. epidermidis* isolates obtained in the first sampling. Additional information includes pulsotype, group to which it belongs, SCC*mec* type, and phenotypic pattern of resistance. The scale at the top indicates the similarity indices (in percentages).

**Figure 3 antibiotics-11-00811-f003:**
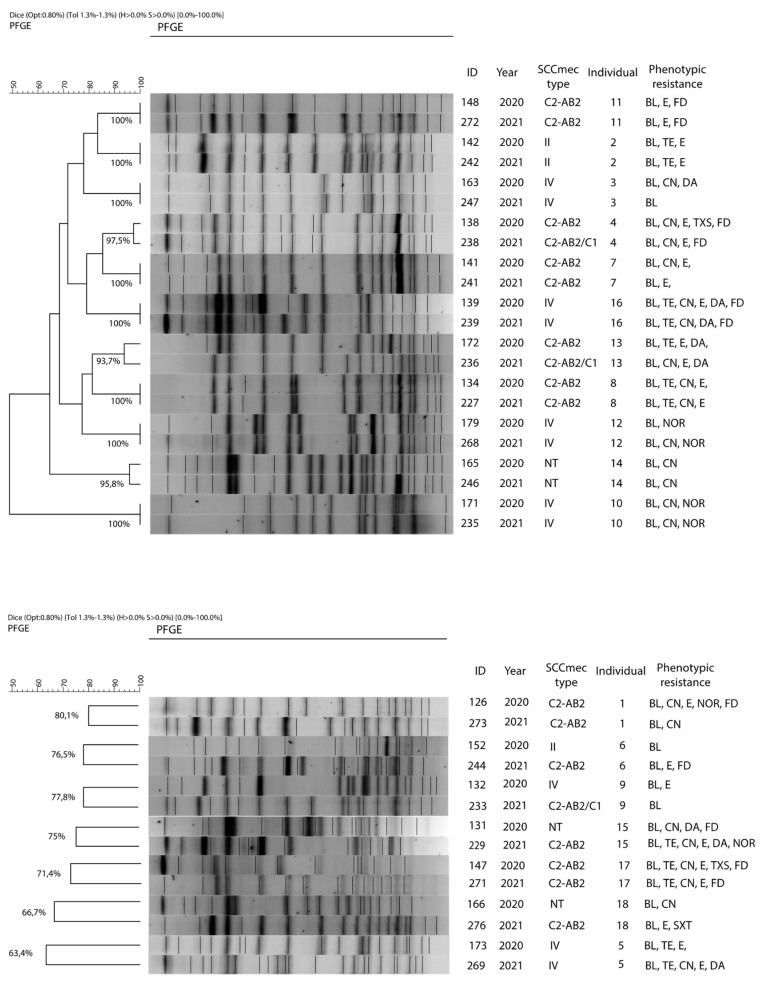
Dendrogram based on PFGE macrorestriction pattern of MR *S. epidermidis* isolates obtained in both samples from each of the persistent carriers. The upper dendrogram shows individuals with “indistinguishable” or “closely related” isolates, and the lower dendrogram shows those with “unrelated” isolates. The scale at the top indicates the similarity indices (in percentages).

**Table 1 antibiotics-11-00811-t001:** Prevalence and individual persistence of gen *mecA*.

Individuals	Samples	Prevalence 2020	Prevalence 2021	General Prevalence	Persistent Carriage	Isolates *mec*A+
94	183	53.2% (50/94)	31.5% (28/89)	61.7% (58/94)	19.1% (18/94)	78

**Table 2 antibiotics-11-00811-t002:** Species identified, percentage and year of isolation.

MRS Species	No. of Isolates	No. of Isolates 2020/21 (Percentage)
*S. epidermidis*	72 (92.3%)	48/24
*S. warneri*	3 (3.8%)	1/2
*S. pseudintermedius*	1 (1.3%)	0/1
*S. haemolyticus*	1 (1.3%)	1/0
*S. sciuri*	1 (1.3%)	0/1

**Table 3 antibiotics-11-00811-t003:** No. of pulsotypes (PTs), no. of strains in each of them, and ID of PTs.

No. of PTs (No. of Strains in Each)	ID PTs
25 (1)	1–7, 9, 13, 14, 16–21, 23, 24, 26–32
4 (2)	8, 10, 12, 15
2 (3)	11, 25
1 (4)	22

**Table 4 antibiotics-11-00811-t004:** *mec*/*ccr* complex, SCC*mec* types, and species and no. of isolates associated.

*mec* Complex Class	*ccr* Complex(es)	SCC*mec* Type (Percentage)	Species (No. of Isolates)
A	AB2	II (11.5%)	*S. epidermidis* (8), *S. warneri* (1)
B	AB2	IV (25.6%)	*S. epidermidis* (20)
C1	AB2	UC (9%)	*S. epidermidis* (6), *S. warneri* (1)
C2	AB2	UC (37.2%)	*S. epidermidis* (28), *S pseudintermedius* (1)
C2	AB2, C1	UC (5.1%)	*S. epidermidis* (3), *S. warneri* (1)
C2	C1	V (1.3%)	*S. epidermidis* (1)
NT	NT	NT (10.3%)	*S. epidermidis* (6), *S. haemolyticus* (1), *S. sciuri* (1)

NT: not typeable; UC: unusual combination.

## Data Availability

The data presented in this study are available upon request.

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
