# Peer review of "Clonality and Persistence of Multiresistant Methicillin-Resistant Coagulase-Negative Staphylococci Isolated from the Staff of a University Veterinary Hospital"

_antibiotics, 2022, doi:10.3390/antibiotics11060811_

Round 1

Reviewer 1 Report

Rey et. al., characterized the methicillin resistant coagulates negative staphylococci (MRCoNSO) isolated from healthy staff of university hospital in Spain over the period of two years to assess the importance as a reservoir for antimicrobial resistance and to determine their population structure and evolution. Most of the isolates were S. epidermidis and majority being resistant to Penicillin, Cefoxitin and Oxacillin. The authors did the sampling six months apart. During the second sampling, the prevalence rate decreased dramatically, and authors suggest that it could be due to hotter temperature and reduced humidity during those months. Therefore, it would be good to provide additional information on exact sampling time and information regarding the weather during those time. Some additional concerns are there such as the age and sex distribution of the subjects to rule out there is no impact on the prevalence outcome. Overall, the study provides an important outlook into the antimicrobial resistance patter of MRCoNSO in veterinary settings and have direct contact with the animals infected with these organisms. The study highlights the importance of One-Health in the wake of pandemic and assert the importance of addressing epidemiology from a One-Health perspective.

Author Response

Thank you very much for your review. I will answer your comments point by point to show you how I have incorporated the suggestions you have made in relation to our article with some considerations in this regard.

“Rey et. al., characterized the methicillin resistant coagulates negative staphylococci (MRCoNSO) isolated from healthy staff of university hospital in Spain over the period of two years to assess the importance as a reservoir for antimicrobial resistance and to determine their population structure and evolution. Most of the isolates were S. epidermidis and majority being resistant to Penicillin, Cefoxitin and Oxacillin. The authors did the sampling six months apart. During the second sampling, the prevalence rate decreased dramatically, and authors suggest that it could be due to hotter temperature and reduced humidity during those months. Therefore, it would be good to provide additional information on exact sampling time and information regarding the weather during those time. Some additional concerns are there such as the age and sex distribution of the subjects to rule out there is no impact on the prevalence outcome. Overall, the study provides an important outlook into the antimicrobial resistance patter of MRCoNSO in veterinary settings and have direct contact with the animals infected with these organisms. The study highlights the importance of One-Health in the wake of pandemic and assert the importance of addressing epidemiology from a One-Health perspective.”

  1. “(…)Therefore, it would be good to provide additional information on exact sampling time and information regarding the weather during those time (…)”.

This additional information has been incorporated on page 11 in lines 343-346: "The study was carried out during the months of December 2020 and June 2021, in a location with climatic characteristics typical of a Mediterranean climate with Atlantic influences: temperate and humid in winter and hot and dry in summer".

  1. “ (…) Some additional concerns are there such as the age and sex distribution of the subjects to rule out there is no impact on the prevalence outcome (…). 

These additional concerns have been incorporated into the article on page 11, lines 348-351: "All the personnel were between 25-55 years old, both groups had a similar male to female ratio, did not report to be suffering from pathological processes at the time of sampling and had not been treated with antimicrobials in the six months prior to sampling."

Reviewer 2 Report

Thank you authors for an interesting study. Suggestions as below for consideration please.

Line 331: Suggest to mention detailed timeline of the study period e.g. between Month 2020 and Month 2021.

Line 332: Suggest to mention brief type of animals, e.g. small animals? large animals? mammals? etc., and what kind of brief clinical manifestations of the animals e.g. communicable diseases? to see if this information can be useful for readers.

Line 333: How did the authors know/confirm the participants were healthy? Or apparently healthy?

Line 333: Suggest to define period e.g. past 6 months for the sentence "had not been previously treated with antibiotics". I dont quite think they had never taken any antibiotics before in their lifetime.

Line 336: Explain why nasal swab was chosen as the anatomical site for sampling for the study, as the 'exposure' described in the para above is 'direct contact with animals'. The choice of anatomical site for sampling can influence on the microflora and hence their susceptibility profiles to antimicrobials

Line 82-85: "A total of 94 people ... could not be sampled in the second one" this sentence should be in Materials & Methods and should not repeat in the results.

Figure 1: FOX and OX phenotypically represents methicillin-resistance. Is there any isolate having discrepancies between mecA gene detection and FOX/OX phenotypical resistance? If so, suggest to include in discussion.

Line 206-21): I think the discussion should also include the variation of different animals in contact with different individuals participated in the study, as well as their varying degrees of 'exposure contacts'.

Materials and Methods: suggest to define multi-drug resistance, as a common definition would be resistance against 3 antibiotic 'classes' 

Line 306: 'these clones not only became dominant..." how did the authors determine if the strains are dominant?

Line 321-322: Suggest to include discussion, in the context of community/public health, on those strains that do not persist over time, however can rapidly evolve and may lead to the chance  transient carrier/transmission/transfer.

Regards

Author Response

Thank you very much for your review. I will answer your comments  point by point to show you how I have incorporated the suggestions you have made in relation to our article with some considerations in this regard.

  • Line 331: Suggest to mention detailed timeline of the study period e.g. between Month 2020 and Month 2021.

This additional information has been incorporated on page 11 in lines 343-346: "The study was carried out during the months of December 2020 and June 2021, in a location with climatic characteristics typical of a Mediterranean climate with Atlantic influences: temperate and humid in winter, hot and dry in summer".

  • Line 332: Suggest to mention brief type of animals, e.g. small animals? large animals? mammals? etc., and what kind of brief clinical manifestations of the animals e.g. communicable diseases? to see if this information can be useful for readers.

The types of animal present in the hospital and the procedures carried out are briefly described in the following explanatory paragraph “mainly small animals suffering from medical, surgical or obstetric conditions of varying aetiologies” in page 11, lines 347-348.

  • Line 333: How did the authors know/confirm the participants were healthy? Or apparently healthy?

To obtain this information, each individual was questioned at the time of sampling about their health status and antimicrobial intake in the previous six months. This additional information has been incorporated into the article on page 11, 348-351 lines: "All the personnel were between 25-55 years old, both groups had a similar male to female ratio, did not report to be suffering from pathological processes at the time of sampling and had not been treated with antimicrobials in the six months prior to sampling."

  • Line 333: Suggest to define period e.g. past 6 months for the sentence "had not been previously treated with antibiotics". I dont quite think they had never taken any antibiotics before in their lifetime.

See previous comment.

  • Line 336: Explain why nasal swab was chosen as the anatomical site for sampling for the study, as the 'exposure' described in the para above is 'direct contact with animals'. The choice of anatomical site for sampling can influence on the microflora and hence their susceptibility profiles to antimicrobials.

The nasal swab was chosen as the nasal mucosa is the ideal anatomical site for the establishment and survival of staphylococci present in a given individual due to its unique temperature and humidity characteristics. This nasal colonization occurs regardless of the type of previous contact with the source of infection. Nasal sampling is therefore the safest and most reliable way to investigate the presence of Staphylococcus in healthy individuals, in fact it is also the sampling site in most of the studies reviewed in our study. This would not be the case when investigating pathological processes related to a specific species, in which case the specific area where the alteration is occurring (skin, abscesses, etc.) would need to be sampled.

  • Line 82-85: "A total of 94 people ... could not be sampled in the second one" this sentence should be in Materials & Methods and should not repeat in the results.

Indeed, the paragraph you point out is a repetition of the one in “Material & methods” (page 11, lines 353-356) and has been deleted.

  • Figure 1: FOX and OX phenotypically represents methicillin-resistance. Is there any isolate having discrepancies between mecA gene detection and FOX/OX phenotypical resistance? If so, suggest to include in discussion.

This discrepancy occurred with the only isolate obtained from S. sciuri, which, despite having the mecA gene, does not phenotypically express this characteristic. This fact is related to the presence in S. sciuri homologous to the mecA gene, chromosomally encoded but not located in the SCCmec, which, unlike what occurs in the rest of Staphyloccus species, does not confer resistance to b-lactams. This peculiarity has been reported in the “Discussion” section on page 9, lines 233-237.

  • Line 206-21): I think the discussion should also include the variation of different animals in contact with different individuals participated in the study, as well as their varying degrees of 'exposure contacts'.

To clarify this point, we have included the following paragraph: Other types of individual differences related to the type of animal or the degree of exposure were not considered relevant since they are mostly small animals handled in a comparable way in the different hospital services, so the degree of exposure is similar” (page 9, lines 220-223)

  • Materials and Methods: suggest to define multi-drug resistance, as a common definition would be resistance against 3 antibiotic 'classes' 

Indeed, this definition is included in page 12, lines 406-407 of “Materials and Methods”: An isolate was considered multi-drug resistant when it showed simultaneous resistance to three or more groups of antimicrobials

  • Line 306: 'these clones not only became dominant..." how did the authors determine if the strains are dominant?

We consider “dominant clones” those that are more prevalent in a certain group whilst also retaining some genetic and phenotypic stability, unlike what happens with the rest of the isolates from that group that occur individually and are unstable. A good example of this is in group 5. In this group the dominant clone is 22 (PT 22), present in 4 individuals, who have identical types of SCCmec and similar patterns of phenotypic resistances; while the rest of the PTs present in that group (2, 4, 5, 14, 21, 23) are represented by a single individual with different genotypic and phenotypic profiles.

  • Line 321-322: Suggest to include discussion, in the context of community/public health, on those strains that do not persist over time, however can rapidly evolve and may lead to the chance transient carrier/transmission/transfer.

To emphasize this point, we have incorporated the following paragraph (page 11, lines 333-337):  “The former, due to their abundance, variability and adaptability, could be responsible for the transfer of the resistance observed between the animal and human spheres, with the repercussions that this fact entails in public health; and the latter, more stable and adapted, being responsible for the clinical processes produced”